# Lidar-Based 3D Obstacle Detection Using Focal Voxel R-CNN for Farmland Environment

Jia Qin [1,2], Ruizhi Sun [1,3], Kun Zhou [4], Yuanyuan Xu [1,2], Banghao Lin [1,2], Lili Yang [1,2], Zhibo Chen [1,2], Long Wen [1,2] and Caicong Wu [1,2,*]

1   College of Information and Electrical Engineering, China Agricultural University, Beijing 100083, China
2   Key Laboratory of Agricultural Machinery Monitoring and Big Data Applications, Ministry of Agriculture and Rural Affairs, Beijing 100083, China
3   Scientific Research Base for Integrated Technologies of Precision Agriculture (Animal Husbandry), Ministry of Agriculture and Rural Affairs, Beijing 100083, China
4   Research & Advanced Engineering, AGCO A/S, DK-8930 Randers, Denmark
*   Correspondence: wucc@cau.edu.cn

**Abstract:** With advances in precision agriculture, autonomous agricultural machines can reduce human labor, optimize workflow, and increase productivity. Accurate and reliable obstacle-detection and avoidance systems are essential for ensuring the safety of automated agricultural machines. Existing LiDAR-based obstacle detection methods for the farmland environment process the point clouds via manually designed features, which is time-consuming, labor-intensive, and weak in terms of generalization. In contrast, deep learning has a powerful ability to learn features autonomously. In this study, we attempted to apply deep learning in LiDAR-based 3D obstacle detection for the farmland environment. In terms of perception hardware, we established a data acquisition platform including LiDAR, a camera, and a GNSS/INS on the agricultural machine. In terms of perception method, considering the different agricultural conditions, we used our datasets to train an effective 3D obstacle detector, known as Focal Voxel R-CNN. We used focal sparse convolution to replace the original 3D sparse convolution because of its adaptable ability to extract effective features from sparse point cloud data. Specifically, a branch of submanifold sparse convolution was added to the upstream of the backbone convolution network; this adds weight to the foreground point and retains more valuable information. In comparison with Voxel R-CNN, the proposed Focal Voxel R-CNN significantly improves the detection performance for small objects, and the AP in the pedestrian class increased from 89.04% to 92.89%. The results show that our model obtains an mAP of 91.43%, which is 3.36% higher than the base model. The detection speed is 28.57 FPS, which is 4.18 FPS faster than the base model. The experiments show the effectiveness of our model, which can provide a more reliable obstacle detection model for autonomous agricultural machines.

**Keywords:** obstacle detection; LiDAR; point clouds; focal voxel R-CNN; farmland





## 1. Introduction

Autonomous agricultural machines can effectively improve the efficiency of agricultural production and reduce the labor intensity of agricultural practitioners [1]. However, there is still a need for a human operator to monitor the environment and respond to potential obstacles and hazards during the operation in a timely manner. In order to fully automate agricultural machines, accurate and reliable obstacle detection and avoidance systems are essential to alleviate safety concerns. At present, autonomous agricultural machines mainly rely on positioning technology (e.g., global navigation satellite system (GNSS)) to plan their operating paths; efforts are underway to fully automate this process, but safety concerns are currently hindering progress in this regard—especially the sudden appearance of dynamic obstacles [2], such as humans crossing the road or other agricultural machines operating in the field. Therefore, in sensing the surrounding environment of

autonomous agricultural machines, the detection and localization of such obstacles is a crucial step.

Currently, research on obstacle detection for autonomous agricultural machines is receiving increasing attention [3]; such studies mainly focus on computer vision [4–6] and LiDAR [2,7,8]. Compared to 2D images, LiDAR can better perceive the surrounding environment, as it generates 3D point cloud data with rich geometric, shape, and scale information. In addition, it is not susceptible to light and, thus, it has better reliability. In agriculture, LiDAR is used to detect the category and location of obstacles to ensure the safety of autonomous agricultural machines during their operation [9]. In practice, 3D point cloud processing faces challenges, such as the unstructured and sparse nature of the point cloud data. The accuracy of traditional machine learning workflows no longer meets the needs of autonomous agricultural machines' operation [9]. Because such methods are feature-based—which means that the point features require manually designed parameters—it often takes a lot of time to extract effective features, and as the environmental complexity increases, the detection accuracy decreases considerably.

As 2D object detection using deep learning methods has achieved remarkable success, many 2D-image-based detectors have been applied in point cloud data-processing for 3D object detection [10]. The most commonly used methods can be classified into two categories: voxel-based methods [11,12], and point-based methods [13,14]. Voxel-based methods convert sparse point clouds into regular representations, such as 3D voxels or 2D bird's eye views (BEVs), which are later processed using 3D or 2D convolutional neural networks to learn point cloud features; these are more efficient in feature extraction, but voxelization inevitably leads to the loss of some location information, resulting in lower accuracy. Point-based methods extract features directly from the point clouds, generally using PointNet [15], which takes the original point cloud as an input and extracts the point set features via iterative sampling and grouping. These methods can better retain accurate location information, but they have a higher computational complexity, leading to a slow detection speed. The 3D obstacle detection using deep learning with 3D point clouds is still in its infancy in the domain of autonomous agricultural machines. To the best of our knowledge, no similar research has been conducted for autonomous agricultural machines. In this study, for the first time, we attempted to conduct 3D obstacle detection for autonomous agricultural machines with point cloud data.

We chose Voxel R-CNN [16], which is a high-performance voxel-based 3D object detector, as a baseline; it takes 3D sparse CNNs as its backbone for feature extraction, and then it uses a Voxel RoI Pooling module to take full advantage of voxel features by using the nearest neighbor perceptual properties of the voxels to extract the neighboring voxels' features. It is notable that the Voxel RoI Pooling module does not need pointwise information, enabling the detector to perform faster than previous state-of-the-art detectors (e.g., PVRCNN [14]); hence, it can better balance efficiency and accuracy.

However, it uses 3D sparse CNNs for feature extraction, but such 3D sparse CNNs treat every feature equally, ignoring the fact that obstacles and background points have a different importance and that distance affects the sparsity of the point cloud. In agricultural scenes, the background is more widespread compared to obstacles, and the posture of pedestrians in the field is uncertain, making obstacle detection more difficult. To improve this, we used a focal sparse convolutional network [17] instead of the original sparse CNN to assign a different importance to each feature during the convolution process and increase the proportion of valuable features, improving the detection accuracy of small, distant objects while removing a large number of background voxels and improving the final detection without requiring too much computational effort to achieve high representation capability and efficiency.

This paper focuses on improving the precision and robustness of 3D obstacle detection for autonomous agricultural machines to better ensure their safety. We improved Voxel R-CNN to increase the detection performance of small, distant objects while ensuring overall accuracy. To test the performance of the model, a solution was presented from data

acquisition and preparation of datasets for model training. Two representative obstacles—pedestrian and tractor—were chosen to conduct different comparison experiments. The main contributions of this study are as follows:

(1) We first attempted to combine a deep learning method with point clouds to perform 3D obstacle detection for autonomous agricultural machines, using its powerful feature-learning capability to generate more robust obstacle-perception models;

(2) We proposed a Focal Voxel R-CNN network with a focal, sparse convolutional neural network instead of 3D sparse CNN, enhancing the learning of valuable features to improve the accuracy of the model;

(3) A multi-modal dataset for 3D obstacle detection for autonomous agricultural machines was constructed using the KITTI Vision Benchmark, which represents a wide range of real agricultural environments containing static and dynamic obstacles in multiple scenarios. Finally, the effectiveness of the proposed method was evaluated using the collected datasets, and the results show that our model outperformed the other 3D object-detection network.

## 2. Materials and Methods

### 2.1. Data Acquisition and Preprocessing

#### 2.1.1. Platform Setup

A JD1204 tractor was used as the acquisition platform, and it was modified and retrofitted to install each hardware device used for the data collection on the tractor. Using the original structure of the tractor, such as the head counterweight ends of the screw holes, aluminum profiles were added to build the sensor bracket. The acquisition system consisted of four parts: (1) a tractor position acquisition system; (2) a LiDAR system; (3) a vision system; and (4) a data processing system. The installation position of each hardware device is shown in Figure 1.

The tractor's position acquisition system uses a CHC CGI-610 GNSS/INS integrated sensor, which combines satellite positioning and inertial measurement to provide high-precision carrier position, altitude, and speed information in real time to meet the needs of long-term, high-precision, and high-reliability positioning applications for autonomous agricultural machines. The acquisition frequency is 100 Hz.

The LiDAR system uses the Velodyne VLP-16 3D LiDAR. Its laser emission beams consist of 16 lines, and each laser beam can provide distance information and reflectivity information. The vertical field of view is $30°$ ($+/-15°$), the horizontal field of view is $360°$, ~300,000 points of data output per second can be acquired, the maximum distance measurement is 100 m, and the acquisition frequency is 10 Hz.

The vision system uses a FLIR BFS-PGE-23S3C model color camera; its resolution is $1920 \times 1200$, the maximum frame rate is 53 FPS, and the acquisition frequency is 20 Hz.

The data processing system uses a Nuvo-810GC IPC with the Ubuntu 18.04 operating system, which is installed in the cockpit and connected to a display to visualize the collected data. Each of the three sensor modules is connected to the IPC in different ways: GNSS/IMU combined navigation through the RS-232 serial port, and LiDAR and camera through an Ethernet protocol for data transfer with the IPC.

#### 2.1.2. Data Acquisition

The experimental data collection was carried out in June 2021 in fields in Miyun District, Beijing. The ROS (Robot Operating System) system was built on an IPC with Ubuntu 18.04 operating system installed. The data of each sensor were recorded in the form of a bag under the ROS system, i.e., which is a file format for storing ROS message data named by its bag extension. Figure 2 shows a frame from the bag displayed using the RVIZ visualization tool, with the image on the left and the corresponding point-cloud data on the right. The collected bag file can be saved to the IPC, after which the point cloud (i.e., pcd file) and the image (i.e., jpg file) can be extracted separately from the bag through the ROS operation command. In the actual tractor-operating scenarios—for

example, the plowing, planting, and harvesting processes—the tractor is equipped with different equipment and enters the field from the agricultural road, so these two scenes were selected for experimental data collection.

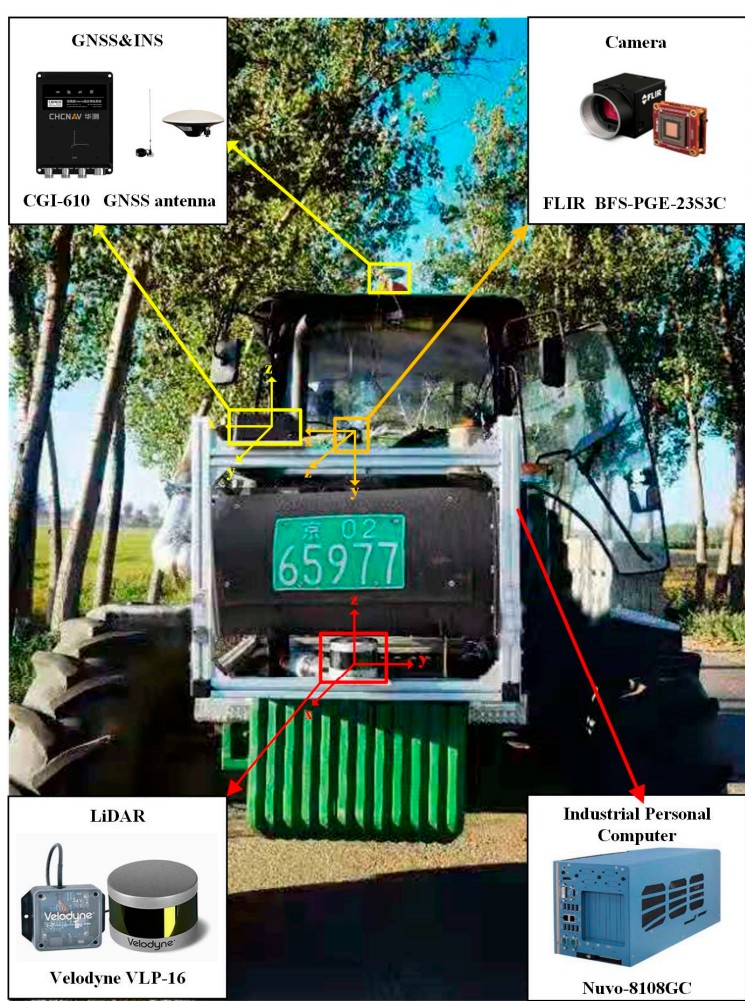

**Figure 1.** Installation diagram of data acquisition system.

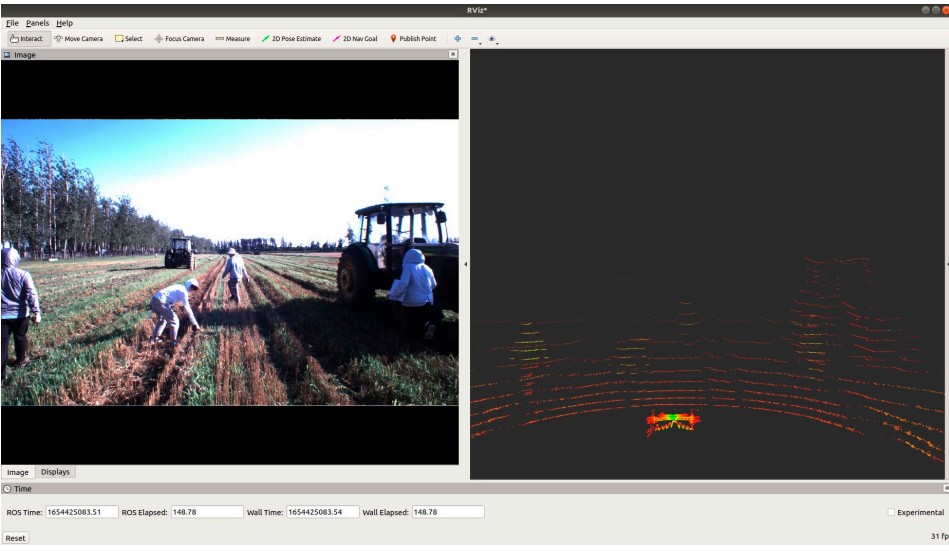

**Figure 2.** RVIZ visualization of the collected data.

### 2.1.3. Data Preprocessing

Since the acquisition frequency and coordinates of each sensor are different, to make the multisource data consistent, it was necessary to carry out the temporal and spatial synchronization of the sensor data, and the preprocessing of the collected data was a preliminary step in the construction of the dataset.

- Temporal synchronization:

Temporal synchronization allows multiple sensors of different frequencies to be synchronized to a uniform frequency. It can also be divided into hard and soft synchronization. Hard synchronization refers to the use of hardware triggers to directly activate multiple sensors through physical signals so that multiple sensors can trigger for sampling at the same moment. Soft synchronization entails synchronizing the data of each sensor to the unified timestamp for each frame of data. In this study, we adopted the soft synchronization method for temporal synchronization due to feasibility concerns. By using the communication mechanism of the ROS, we subscribe to the topic of the camera and LiDAR sensor—which store the corresponding image and point cloud data, respectively—and encapsulated the data as messages for delivery. When a frame of LiDAR point cloud data was received, we would find the nearest image data for matching. Finally, after encapsulating the synchronized point cloud and image data into messages, we published them with different topics and then recorded the new time-synchronized data through the ROS bag, which is the primary mechanism for data logging in the ROS, thereby completing the temporal synchronization for different frequency sensors.

- Spatial synchronization:

Spatial synchronization uses the coordinate transformation relationship between different sensors to convert them to the same coordinate system. In this study, we adopted an offline calibration tool to calibrate the sensor, using Autoware's open-source calibration toolkit to calibrate the LiDAR and camera. The camera's intrinsic matrix was calculated first, after which we calculated the camera-to-LiDAR extrinsic matrix. The intrinsic matrix represents the transformation relationship between the camera coordinate system and the pixel coordinate system, which is built into the camera settings and can be obtained using the aforementioned open-source tools. The extrinsic matrix consists of a rotation matrix and a translation matrix. First, we calculated the rotation matrix of the camera and the LiDAR, and then we optimized the translation matrix. The dimensions of the two matrices are as follows:

$$\mathbf{R}_{velo}^{cam} \in \mathbf{R}^{3\times3} \tag{1}$$

$$\mathbf{t}_{velo}^{cam} \in \mathbf{R}^{1\times3} \tag{2}$$

After the intrinsic calibration, the position and orientation of each checkerboard in the camera coordinate system can be obtained, while the grasping point corresponding to each checkerboard is extracted from the LiDAR point cloud. Therefore, the optimization goal is to align these planes by adjusting the camera's extrinsic parameters, and the rotation translation matrix between sensors is obtained after iterative optimization; this matrix is

$$\mathbf{T}_{velo}^{cam} = \begin{pmatrix} \mathbf{R}_{velo}^{cam} & \mathbf{t}_{velo}^{cam} \\ 0 & 1 \end{pmatrix} \tag{3}$$

Finally, a 3D point $\mathbf{x} = (x, y, z, 1)^T$ in the LiDAR coordinate system is projected to the point $\mathbf{y} = (u, v, 1)^T$ in the 0th camera image, with the following formula:

$$\mathbf{y} = \mathbf{P}_{rect}^{(i)} \mathbf{R}_{rect}^{(0)} \mathbf{T}_{velo}^{cam} \mathbf{x} \tag{4}$$

where $\mathbf{P}_{rect}^{(i)}$ is the projection matrix of the camera, as follows:

$$\mathbf{P}_{rect}^{(i)} = \begin{pmatrix} f_u^{(i)} & 0 & c_u^{(i)} & -f_u^{(i)} \\ 0 & f_v^{(i)} & c_v^{(i)} & 0 \\ 0 & 0 & 1 & 0 \end{pmatrix} \tag{5}$$

where $\mathbf{R}_{rect}^{(0)}$ is the corrected rotation matrix, which is set as a unit matrix, leaving the image uncropped in size.

### 2.2. Dataset

This paper refers to the driving of autonomous vehicles; this kind of application requires real-time object detection to ensure safe driving, which is consistent with the need for the sensing of obstacles by autonomous agricultural machines in farming work. By referring to the standards provided by the KITTI Vision Benchmark Suite [18]—a benchmark dataset for autonomous vehicle driving—we generated a 3D obstacle-detection dataset for agricultural scenarios. After the data preprocessing operation in Section 2.1.3, the temporally and spatially synchronized multisource data were obtained. Then, the annotation of images and point clouds was performed using the Bayside annotation software, and the fusion annotation-required camera parameters, which could be obtained from the aforementioned calibration results of the LiDAR and camera. As shown in Figure 3, the labeling was based on the calibration parameters, and the point cloud was labeled with a 3D box (Figure 3a), which could be automatically mapped to the 2D image (Figure 3b) to generate a rectangular box. These 2D images were later processed to generate labels, and they were also better for visualizing the labeling results.

The final labeling result is given in JSON file format, including the center point of the 3D box; the length, width, height, and rotation angle of the 3D box; the type of obstacle information; and the coordinates of the upper-left and lower-right corners of the 2D box. Finally, the JSON file of the labeling results is rewritten to a txt file in KITTI format using Python script. As shown in Table 1, the format of the dataset labels was set with nine values, including the 2D bounding box information for the final visualization, and the class types were set to Tractor and Pedestrian according to the importance and probability of an obstacle's appearance. A total of 1450 frames of point cloud samples were obtained and divided into training, validation, and test sets at an 8:1:1 ratio.

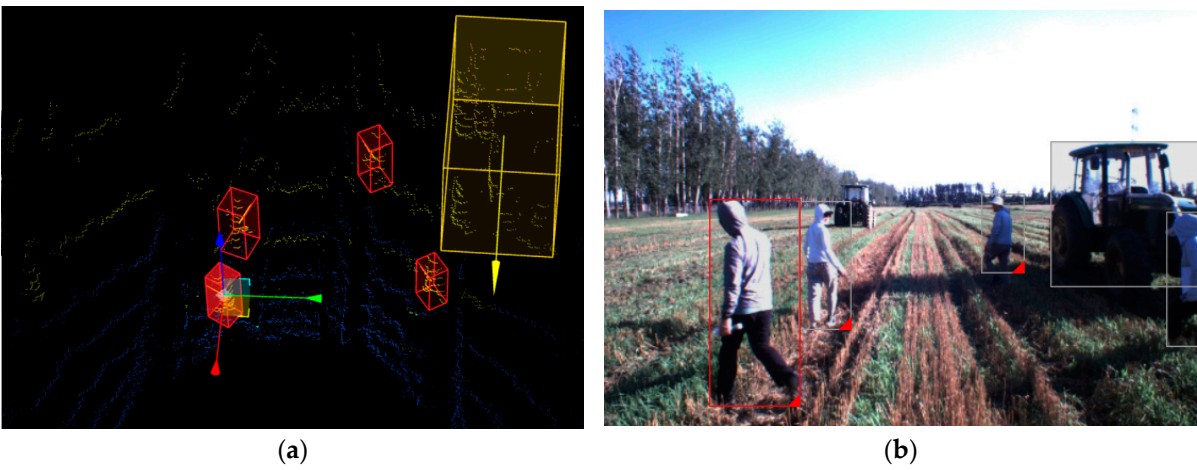

　　　　　　　　(**a**)　　　　　　　　　　　　　　　　　　　　　　　(**b**)

**Figure 3.** Examples of labeling operations. (**a**) Point cloud labeling; (**b**) Image labeling.

As shown in Figure 4, the labeled ground-truth boxes were displayed visually using MATLAB (2022, MathWorks Inc., Natick, MA, USA). The figure shows that two types of obstacles are labeled—namely "tractor" and "pedestrian"—which are projected onto the image from the point cloud in order to facilitate the visualization of the 3D ground-truth boxes.

**Table 1.** Dataset label format.

| Label Parameter | Explanation |
| --- | --- |
| Type | Tractor, Pedestrian |
| Truncation degree | 0 (untruncated)–1 (truncated) |
| Occlusion rate | Degree of occlusion (0 means no occlusion) |
| Observation angle | Observation angle of the object, range: $-\pi \sim \pi$ |
| 2D bounding box | Coordinates of the upper-left and lower-right corners of the 2D box |
| 3D box size | 3D box height, width, and length |
| 3D box position | Coordinates of the 3D box's center in the camera coordinate system |
| Target orientation | Spatial orientation of the 3D object |
| Score | Confidence in the training results used to evaluate the detection performance |

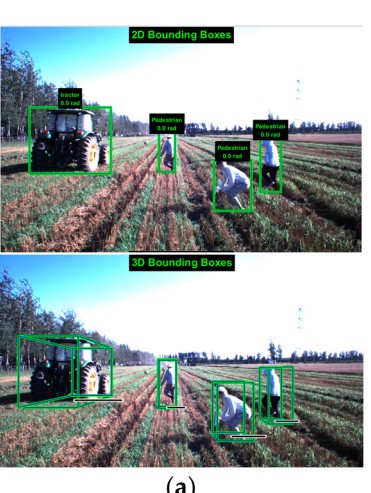 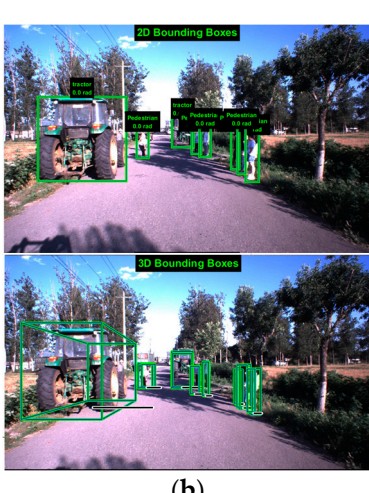

**Figure 4.** Visualization of annotated ground truth: The first row is the annotated 2D ground truth, and the second row is the projection of the annotated 3D ground truth onto the image. (**a**) Field scene; (**b**) Agricultural road scene. (**a**) shows the ground truth of obstacle labeling in the field scene, while (**b**) shows the ground truth of obstacle labeling in the agricultural road scene.

The agricultural road scene mainly records the data of tractors stopping at the roadside and pedestrians crossing the road dynamically; the field scene mainly records the data of multiple tractors operating in the different field waylines and pedestrians crossing the farming field dynamically in multiple postures.

### 2.3. Focal Voxel R-CNN

The Focal Voxel R-CNN is designed based on Voxel R-CNN and we modified its architecture to fit farmland conditions. The original Voxel R-CNN and the proposed Focal Voxel R-CNN are both voxel-based two stage 3D object detectors. As shown in Figure 5, they share a similar framework.

In the first RPN stage, the input point clouds are voxelized into voxels, then these voxelized inputs are gradually converted into volumetric features by the 3D backbone network. After that, BEV feature maps are produced by stacking the 3D volume feature along the Z axis. In the 2D backbone network, features are extracted from the BEV pseudo-images, and then proposals are generated. In the second RCNN stage, the Voxel RoI Pooling is used to extract RoI features, which are then fed into the detect head for prediction box refinement. In Algorithm 1, we show the complete pseudocode of our method.

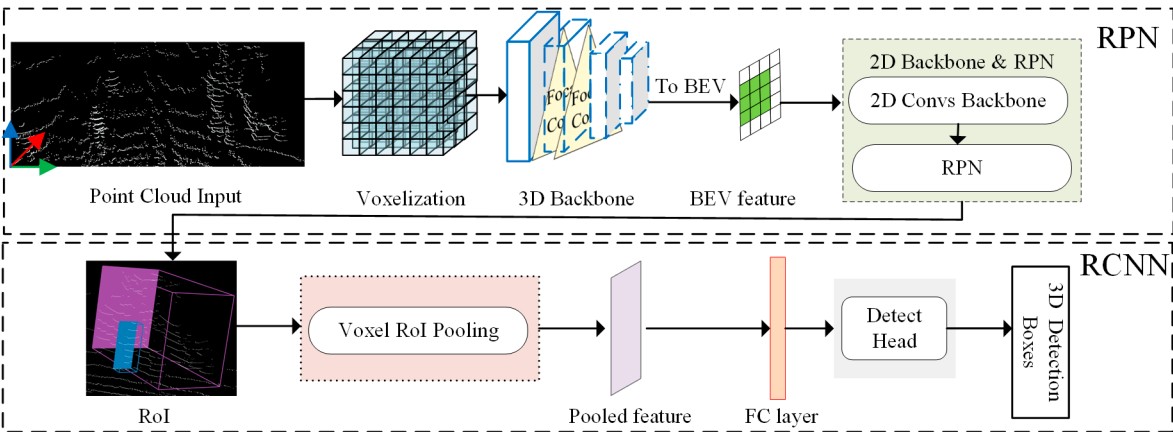

**Figure 5.** The structure of the Focal Voxel R-CNN.

---

**Algorithm 1** Focal Voxel R-CNN training procedure

---

**Inputs:** Dataset $D$, batch size $B$, iteration maximum $N$, model $M$ parameterized by $\beta$, learning rate $\alpha$.
**Outputs:** updated model $M$ with new parameter $\beta^*$

1:  **for** n = 1 to maximum iteration $N$ **do:**
2:   Take one batch of point cloud files $P$ from dataset $D$
3:   Voxelize the point cloud as $P^*$
4:   Extract 3D volume feature $F$ from $P^*$ by 3D backbone
5:   Compress the height of $F$ into BEV form
6:   Extract 2D feature $F^b$ by 2D backbone
    Generate proposals and scores from $F^b$ by RPN
7:   Select the top 128 proposals by scores
    NMS(threshold = 0.7) -> top 100 proposals $S$
8:   Extract RoI feature $F^R$ from $S$ and $F$ by Voxel RoI Pooling
    Predict refined 3D bounding boxes and confidence scores from $F^R$ by Detect head
9:
    NMS(threshold = 0.1) -> final results
10:   Compute loss with Eq 12.
11:   Update the parameter $\beta$
12:  **end for**
13:  **Return** updated model $M$ parameterized by $\beta^*$

---

  We will discuss the detailed architecture of the Focal Voxel R-CNN in the following sections:

1. Voxelization:

  In order to extract features from the sparse point cloud, a 3D space is divided into equally spaced voxels along $x$, $y$, and $z$ axes. In this way, we are able to use the regular voxel as an input for the 3D convolution feature extraction. The original feature of each point is represented by a four-dimensional vector with its $x$, $y$, and $z$ coordinates and intensity. Each non-empty voxel is represented by the mean value of the point cloud feature.

2. The 3D backbone:

  The 3D backbone network usually uses sparse convolution [17] to abstract non-empty voxels into 3D feature volumes via convolutional multiplication operations; as shown in Figure 6a, there are typically one stem and four stages layer in a 3D backbone. Except for the first stage, each stage includes a regular sparse convolution for downsampling and two submanifold blocks. The Subm block is a conv-bn-relu layer, its structure is shown in Figure 6d, which consists of one submanifold sparse convolution followed by batchnorm and ReLU activation. The Reg block's structure is similar to the Subm block, except for the submanifold sparse convolution being replaced by the regular sparse convolution.

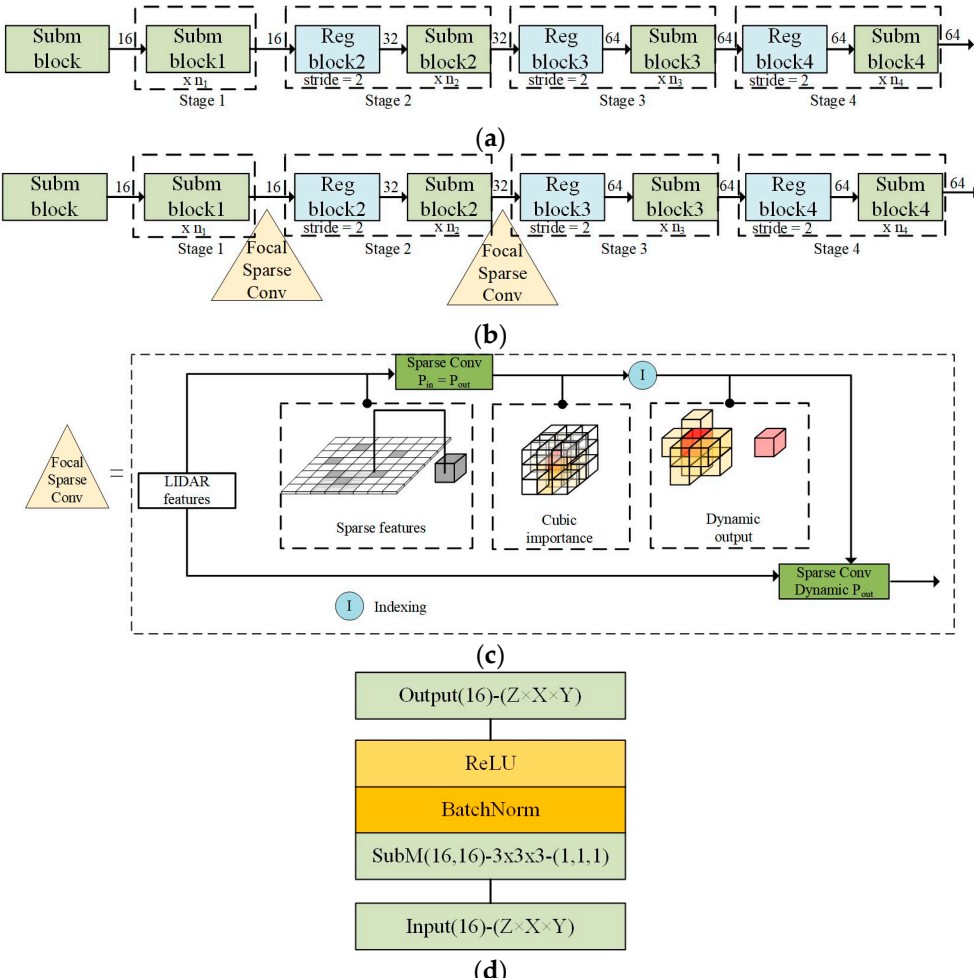

**Figure 6.** (**a**) Structure of the original 3D backbone. (**b**) Structure of the modified 3D backbone. (**c**) Structure of a focal sparse convolutional network. (**d**) Structure of the Subm block.

In our work, we added the focal sparse convolutional neural network [17] in the last layer of Stages 1 and 2 based on the original network, which can make the feature learning process focus on more valuable foreground point data. As shown in Figure 6c, the focal sparse convolutional network adds a branch of submanifold sparse convolution upstream of the main convolution to dynamically output the shapes of features with different importance. It uses the predicted cubic importance to dynamically determine the output shape of the input features. It mainly includes three steps: first, the cubic importance map $I^p$ is constructed by making dynamic predictions for each input feature through the sigmoid function, which includes the importance scores of the output features around the input features at position p. Next, the important input features are selected as shown in Equation (6), where $P_{im}$ is a subset of the input features ($P_{in}$) containing relatively important input features, $I_0^p$ is the center of the importance cube map at position p, and $\tau$ is a predefined threshold, where this equation becomes a regular or submanifold sparse convolution when it is 0 or 1, respectively. Finally, a dynamic output shape is generated, in which the $P_{im}$ features are reshaped according to the predefined threshold $\tau$, and the final output shape $K_{im}^d(p)$ is shown in Equation (7).

$$P_{im} = \left\{ p \,\middle|\, I_0^p \geq \tau, p \in P_{in} \right\} \tag{6}$$

$$K_{im}^d(p) = \left\{ k \,\middle|\, p + k \in P_{in}, I_k^p \geq \tau, k \in K^d \right\} \tag{7}$$

As a result, the network's downsampling process gradually increases the valuable information, improving the recognition accuracy of small objects while removing a large number of background voxels and improving the final detection without causing too much computational burden. The modified 3D backbone structure is shown in Figure 6b.

3.  The 2D backbone and Region Proposal Network

After being processed by the 3D backbone, the sparse 3D feature volumes are stacked in the Z-axis and transformed into a bird's-eye view (BEV) form, and then the 2D backbone network extracts 2D features from the BEV pseudo-image feature map. There are two components in the 2D backbone: two blocks of $3 \times 3$ convolution layers, and an upsampled and concatenated multi-scale feature fusion network. Finally, the RPN is applied to generate 3D region proposals.

4.  Voxel RoI Pooling

Voxel RoI Pooling Layer is used to extract RoI features from proposals that are generated in the RPN stage. First, a region proposal is divided up into $6 \times 6 \times 6$ regular sub-voxels. Then, a voxel query is used to integrate the neighboring voxel features. The neighboring voxel features are subsequently aggregated to a more multi-scale manner using an accelerated PointNet Module.

5.  Detect Head

In this step, we will use the RoI features extracted by the Voxel RoI Pooling and feed them into the detect head for box refinement, which will be divided into two branches: one for bounding box regression and the other for confidence prediction.

*2.4. Loss Function*

We trained the proposed network by an end-to-end approach with the region proposal loss $L_{rpn}$ and the proposal refinement loss $L_{rcnn}$. The total loss is calculated by the sum of the region proposal loss $L_{rpn}$ and the proposal refinement loss $L_{rcnn}$, as:

$$L_{total} = L_{rpn} + L_{rcnn} \tag{8}$$

In the first RPN stage, we adopted the same $L_{rpn}$ with [16], which is the sum of classification loss and box regression loss, as:

$$L_{rpn} = \frac{1}{N_{fg}} \left[ \sum_i L_{cls}(l_i, l_i^*) + \sigma \sum_i L_{reg}(\gamma_i^1, \gamma_i^*) \right] \tag{9}$$

where $N_{fg}$ is the number of foreground proposals; $l_i$ represents the prediction value of the classification category; $l_i^*$ represents the target value of the ground truth category. $\sigma$ denotes that only foreground proposals are used to calculate box regression loss. $\gamma_i^1$ indicates the prediction value of box regression; $\gamma_i^*$ indicates the ground truth value of box regression. For classification, we used Focal Loss [19] and for box regression, we used Huber loss.

In the second RCNN stage, the proposal refinement loss was calculated by confidence score prediction loss $L_{iou}$ and box refinement loss $L_{reg}$. In the confidence prediction branch, the target value is an IoU-guided value, as:

$$c_i^* = \begin{cases} 0 & \text{IoU}_i < \theta_L, \\ \frac{\text{IoU}_i - \theta_L}{\theta_H - \theta_L} & \theta_L \leq \text{IoU}_i < \theta_H, \\ 1 & \text{IoU}_i > \theta_H, \end{cases} \tag{10}$$

where $c_i^*$ represents the IoU between the i-th proposal and its ground truth box; $\theta_L$ and $\theta_H$ are thresholds for foreground and background IoU, respectively. In this case, we used Binary Cross Entropy Loss for confidence prediction, and Huber Loss for box refinement; the losses of the RCNN stage are computed as:

$$L_{rcnn} = \frac{1}{N_s}\left[\sum_i Liou(c_i, c_i^*) + \varepsilon\sum_i L_{reg}(\gamma_i{}^2, \gamma_2{}^*)\right] \tag{11}$$

where $N_s$ represents the number of proposals that were sampled during training; $c_i$ represents the prediction of the confidence score; $c_i^*$ represents the ground truth of the confidence score; $\varepsilon$ denotes that only foreground proposals contribute to the box regression loss; $\gamma_i{}^2$ denotes the prediction value of box regression; $\gamma_2{}^*$ denotes the ground truth of box regression.

*2.5. Evaluation Metrics*

The evaluation metrics used in this experiment included recall (R), and mAP score—the same as the KITTI evaluation. The formulae for recall and mAP are as follows:

$$\text{Recall} = \frac{\text{TP}}{\text{TP} + \text{FN}} = \frac{\text{TP}}{\text{all ground truths}} \tag{12}$$

$$\text{mAP} = \frac{1}{C}\sum_{i=1}^{|C|} \text{AP}_i \tag{13}$$

$$\text{AP} = \frac{1}{N}\sum_{r \in S} P(r) \tag{14}$$

$$P(r) = \max_{\tilde{r}:\tilde{r} \geq r} P(\tilde{r}) \tag{15}$$

where TP stands for true positives, which indicates that the model correctly predicts positive cases, i.e., the number of correctly detected tractors or pedestrians; FP stands for false positives, which indicates the number of wrong detections by the model; and FN stands for false negatives, which indicates the number of missed detections. The mean AP (mAP) can be obtained by averaging the AP of all categories in a given dataset. C is the number of categories being detected, in this study we had two categories—namely tractor and pedestrian—so C = equals 2. KITTI adopts the AP@SN metric, and we followed the same metric to evaluate our model. The AP followed the AP@S40 metric [20], so N = equals 40, which means that 40 recall levels were evaluated, where S40 = [1/40, 2/40, 3/40, . . . , 1]. $P(r)$ calculates the area under the precision-recall curve, taking the maximum precision whose recall is greater than the threshold $\tilde{r}$.

**3. Results**

*3.1. Multisensor Spatiotemporal Synchronization Results*

3.1.1. Time Synchronization Results

To verify the results of multisensor time synchronization, the rqt_bag command of the ROS was used to visualize the multisensor timestamps.

Figure 7 shows the comparison of the results of multisensor time synchronization. From Figure 7a, it can be seen intuitively that the two unsynchronized sensor timestamps do not have the same timing correspondence; for example, at 0.0 s, the green vertical line—which represents one frame of image timestamp information—and the purple vertical line—which represents one frame of LiDAR timestamp information—are not consistent, and the timestamp information of the camera appears more frequently within 1 s, indicating that the camera has a higher acquisition frequency.

Meanwhile, after time synchronization (Figure 7b), it can be seen that the timestamps of the two sensors become very close to one another, the frequencies of the two sensors become identical, and both of them are collected at an interval of 0.1 s.

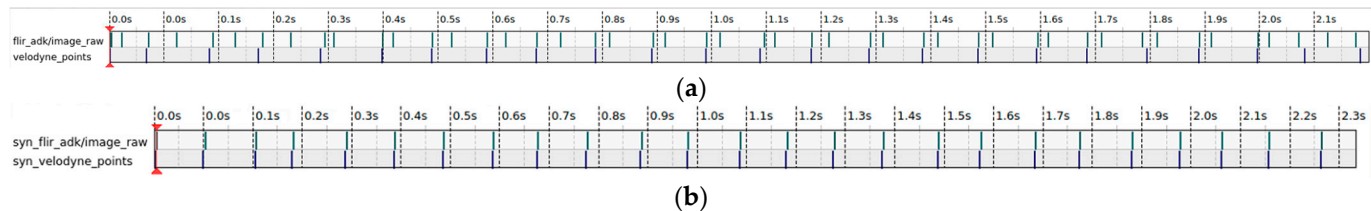

(a)

(b)

**Figure 7.** Comparison of the results of multisensor time synchronization: The horizontal axis represents the time, the vertical axis is the sensor topic name, flir_adk/image_raw represents the image data collected by the camera, and velodyne_points represents the point cloud data collected by the LiDAR. (**a**) Unsynchronized multisensor timestamps. (**b**) Synchronized multisensor timestamps.

3.1.2. Spatial Registration Results

After the calibration of the camera and LiDAR using Autoware's open-source Calibration_Toolkit, the results were validated using a visualization tool.

Figure 8 shows the visualization results between the LiDAR and the camera; the LiDAR line beam is projected onto the image, and the different colored lines represent different reflectance of the object, where it can be seen that the red line projected onto the edge of the black and white calibration checkerboard is flat, while the orange line projected onto the tree also shown in a rough outline, which represents a relatively accurate spatial synchronization result obtained between the LiDAR and the camera.

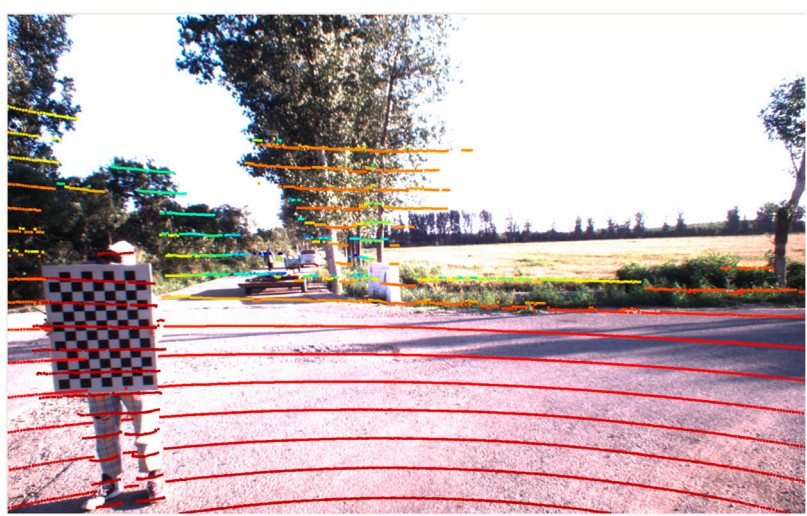

**Figure 8.** Visualization of spatial registration results between the LiDAR and the camera.

*3.2. Comparison between Different Methods*

To validate the model's effectiveness, the improved Voxel R-CNN model was compared with three state-of-the-art models, which comprised two types of representation of point clouds—namely, point-based (PV-RCNN [14]) and voxel-based (SECOND [12] Voxel R-CNN [16]) methods—and used the same training parameters and datasets for training. The results of the experiments comparing different methods for obstacle detection are shown in Table 2.

As shown in Table 2, the experimental results of our method and the comparison methods indicate that our Focal Voxel R-CNN method outperforms the other state-of-the-art methods. For example, in terms of recall, the Focal Voxel R-CNN method has a recall of 90.23%, which is 1.82%, 7.35%, and 1.75% higher than the recall of the original Voxel R-CNN, SECOND, and PV-RCNN methods, respectively. In terms of mAP values, it improved by 3.36%, 15.38%, and 1.76%, respectively. We found that our method enhanced detection performance in both classes with respect to current methods. The AP for the tractor class

increased by 2.86%, 11.88%, and 0.26% in comparison to Voxel R-CNN, SECOND, and PVRCNN, whereas the AP for the pedestrian class also increased considerably by 3.85%, 18.88%, 3.26%. Although our model's memory increased by 2.4% compared to the original Voxel R-CNN model, the frame rate was 4.18 FPS faster in comparison, the single-frame detection speed is 0.03 s, better balancing the detection accuracy and efficiency. In Figure 9, we can see that our method converges better than the original Voxel R-CNN, and the final loss is much lower than the original Voxel R-CNN.

**Table 2.** Comparative experimental results of different obstacle detection methods.

| Method | R (%) | AP (%) | | mAP (%) | Speed (FPS) | Model Size (MB) |
| | | Tractor | Pedestrian | | | |
| --- | --- | --- | --- | --- | --- | --- |
| PV-RCNN | 88.48 | 89.70 | 89.63 | 89.67 | 17.67 | 150.2 |
| SECOND | 82.88 | 78.08 | 74.01 | 76.05 | 42.55 | 60.9 |
| Voxel R-CNN | 88.41 | 87.10 | 89.04 | 88.07 | 24.39 | 105.3 |
| Our method | 90.23 | 89.96 | 92.89 | 91.43 | 28.57 | 107.9 |

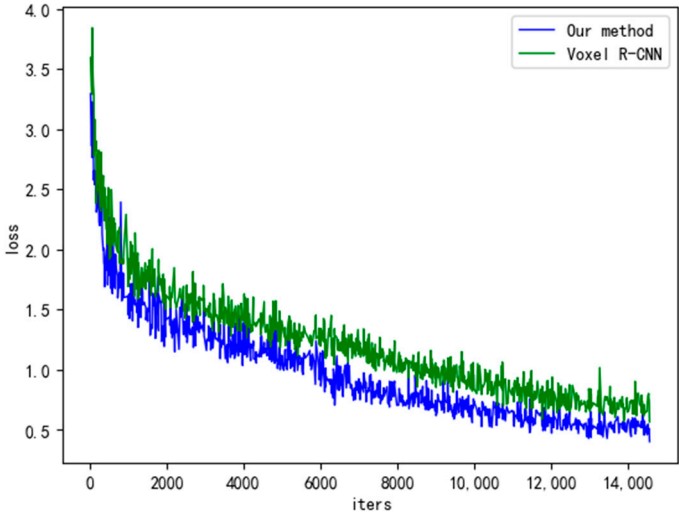

**Figure 9.** Loss curve of Voxel R-CNN and our method.

*3.3. Comparison between Different Voxel Sizes*

To test the robustness of the model, different voxel sizes were selected for comparison experiments, and the experimental results are shown in Table 3.

**Table 3.** Comparison of model detection results under different voxel sizes.

| Voxel Size (m) | Method | R (%) | mAP (%) | Speed (FPS) | Model Size (MB) |
| --- | --- | --- | --- | --- | --- |
| (0.05, 0.05, 0.1) | PV-RCNN | 88.48 | 89.67 | 17.67 | 150.2 |
| | SECOND | 82.88 | 76.05 | 42.55 | 60.9 |
| | Voxel R-CNN | 88.41 | 88.07 | 24.39 | 105.3 |
| | Our method | 90.23 | 91.43 | 28.57 | 107.9 |
| (0.1, 0.1, 0.2) | PV-RCNN | 82.28 | 84.21 | 31.34 | 150.2 |
| | SECOND | 79.04 | 69.09 | 84.75 | 60.9 |
| | Voxel R-CNN | 84.43 | 84.63 | 31.25 | 105.3 |
| | Our method | 86.12 | 89.11 | 37.45 | 107.9 |

As can be seen from Table 3, our model still performs well with voxel sizes of (0.1, 0.1, 0.2) m, and the mAP value is 89.11%, which is an improvement of 4.9%, 20.02%, and 4.48% compared to the PV-RCNN (84.21%), SECOND (69.09%), and Voxel R-CNN (84.63%)

methods, respectively. Overall, the accuracy decreases when the size of the divided voxels becomes larger, and the model's memory remains unchanged, while the detection speed is improved. The detection results of our model for two different voxel sizes are improved compared with the other methods, indicating that our model has strong robustness.

### 3.4. Visualization of Obstacle Detection Results

In order to qualitatively analyze the Focal Voxel R-CNN method, experiments were carried out to visualize the detection results in different scenes, and some of the obstacle detection visualization results are shown in Figure 10.

From Figure 10, it can be seen that when pedestrians and tractors appear in the point cloud, the generated 3D bounding box is wrapped around the obstacle. The color of the 3D bounding boxes for the tractor category is green, while for the pedestrian category, they are blue, and the red boxes are the ground truth. From the point-cloud detection results in Figure 10, it can be seen that the overlap between the detection boxes and the ground-truth boxes is high, which indicates that our proposed method can accurately detect the obstacles.

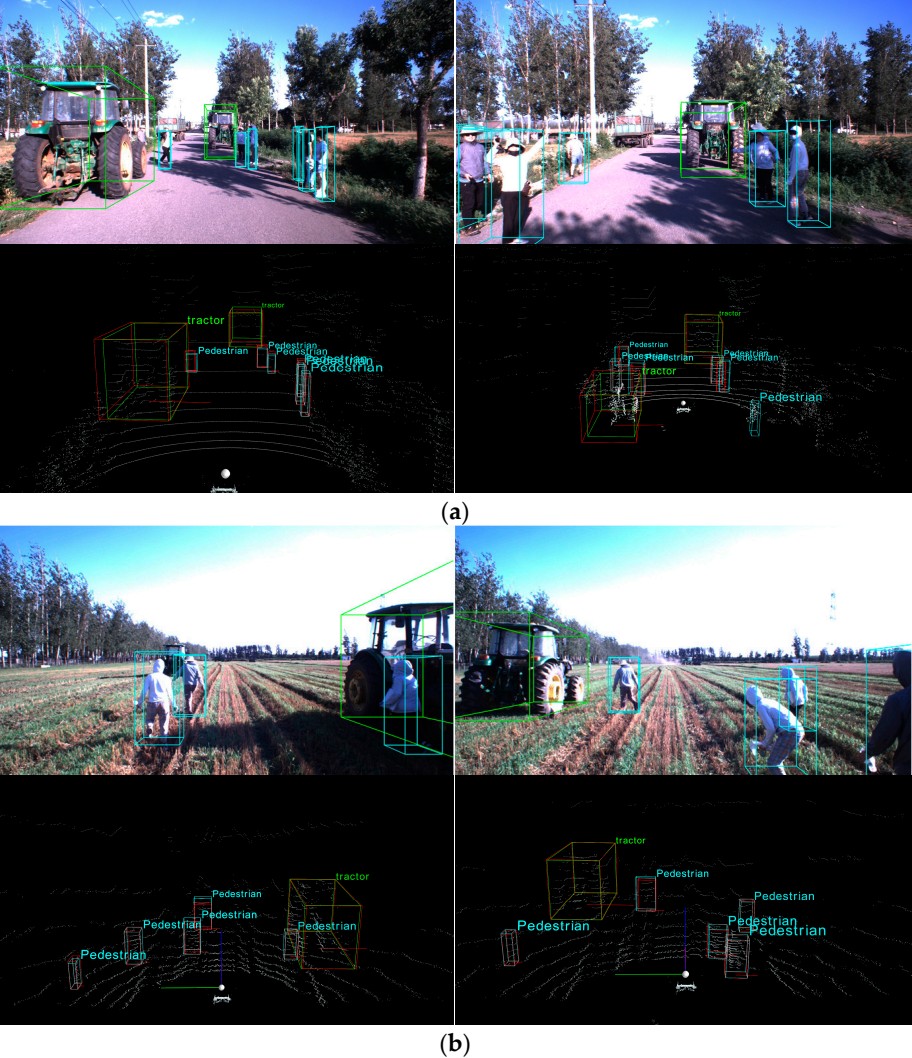

**Figure 10.** Visualization of obstacle detection results in different scenes. For each sample, the upper part shows the 3D detection boxes projected to the image, while the lower part shows the 3D detection results from the point cloud. (**a**) Agricultural road scene. (**b**) Field scene.

## 4. Discussion

In this study, a multi-sensor acquisition device was established and mounted on a tractor to collect data for agricultural obstacle detection. A spatiotemporal synchronized dataset was built to implement deep learning for 3D point clouds, and a focal sparse CNN was added to enhance the extraction of effective features from the original Voxel R-CNN network. The proposed Focal Voxel R-CNN model achieved good results for obstacle detection on a point cloud dataset containing pedestrians and tractors, showing the potential to be able to provide safe and effective sensing information for autonomous agricultural machines, e.g., as an input to path planning for obstacle avoidance.

### 4.1. Analysis of Results

Based on the detection results in Table 2, the accuracy and speed of the method are not easily balanced. Normally, the higher the accuracy, the more features need to be extracted and the slower the speed. For example, SECOND achieves the highest speed (which is 42.55 FPS), but its mAP is only 76.05%. In contrast, PVRCNN has higher accuracy; its mAP is 89.67%, but its speed is considerably slow, at only 17.67 FPS. Our Focal Voxel R-CNN achieves the best results, with a recall of 90.23%, and a mAP of 91.43%, indicating that the addition of the focal sparse convolutional network concentrates the extraction of features to more valuable foreground points. It is noteworthy that the AP in the *pedestrian* class increased from 89.04% to 92.89%. This indicates that the selection of features based on the importance cube helps to retain more valuable features and, ultimately, improves the detection accuracy of the small object. Our model's detection speed is 28.57 FPS, which is 4.18 FPS faster than the base model. The corresponding model size was also increased as expected, but the increase was small—from 105.3 MB to 107.9 MB. Meanwhile, different sizes of voxel divisions were selected; specifically, we used the common division size of (0.05,0.05,0.1) m and doubled it to (0.1,0.1,0.2) to test the robustness of the model. Based on the test results in Table 3, the accuracy is also affected when the voxel size increases from (0.05,0.05,0.1) m to (0.1,0.1,0.2) m, and the mAP decreases from 91.43% to 89.11%. This is because when the voxels become larger, there will be more point clouds in each voxel, and some locally effective information may be lost during the feature extraction process. However, at the same time, the detection speed of the model becomes faster as the voxel size increases. This is due to the overall reduction in the number of voxels input to the 3D backbone network for convolutional operations, so the processing time is reduced accordingly, and the detection speed of the model increases from 28.57 FPS to 37.45 FPS.

### 4.2. Analysis of Failure Cases

Typical obstacles that fail to be detected correctly are shown in Figure 11.

Occlusion and multiple poses of the detected obstacles cause difficulty in extracting features, resulting in failed obstacle detections. On the one hand, the occlusion between obstacles—such as the overlap of moving pedestrians—causes the LiDAR's laser beams to fail to reach the occluded objects, leading to the sparseness of the obtained point cloud data, which makes the feature extraction process more difficult and, ultimately, leads to missed detection or reduced accuracy. On the other hand, the protruding limbs of some pedestrians with multiple postures can be mistakenly detected as pedestrians. This is mainly because the pedestrian itself is a small target and the point cloud data are sparse, so the learned features are not very distinguishable, and it is easy to mistakenly detect some other relatively small targets for pedestrians—such as human thighs—leading to the false recognition of irrelevant information. It is worth noting that although only two types of obstacles (i.e., pedestrian and tractor) have been tested so far, we have established a method to detect specific categories of obstacles for autonomous agricultural machines, and this method could be applied to more types of obstacles if necessary.

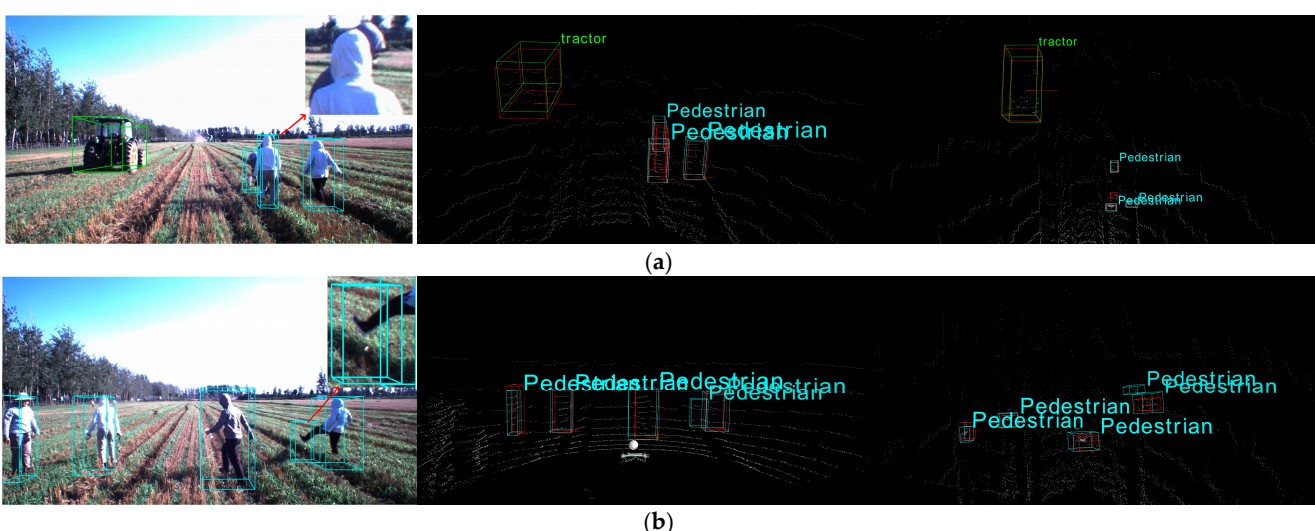

**Figure 11.** Typical failed detection cases. (**a**) Under occlusion. (**b**) Small targets in multiple poses.

## 5. Conclusions

In conclusion, we presented a LiDAR-based 3D obstacle detector known as Focal Voxel R-CNN for farmland environment. First, we built a multi-sensor acquisition device on a tractor to collect data from multiple sensors. Then, a spatiotemporally synchronized dataset was constructed to implement deep learning for 3D point clouds. Additionally, a focal sparse CNN was added to the 3D backbone to enhance the ability to extract effective features from the data. Finally, the comparison experiments show the effectiveness of our modifications and that our improved model can outperform other state-of-the-art object detectors on our dataset, showing a higher detection accuracy and better robustness, which can provide a more reliable obstacle detection model for autonomous agricultural machines.

In future work, we will consider adding more types of obstacles found in farming operations to the existing dataset and utilizing the image information from the camera for multimodal fusion to further improve the accuracy of the obstacle detection model.

**Author Contributions:** Conceptualization, validation, and writing—review and editing, J.Q. and K.Z.; methodology, software, formal analysis, investigation, writing—original draft preparation, J.Q.; resources and data curation, J.Q., Y.X., L.Y., Z.C., B.L. and L.W.; supervision, R.S. and C.W.; project administration, C.W.; funding acquisition, C.W. All authors have read and agreed to the published version of the manuscript.

**Funding:** This research was funded by the Beijing Municipal Science and Technology Project, grant number Z201100008020008. This research was also funded by National Key Research and Development Project, grant number 2021YFB3901302.

**Institutional Review Board Statement:** Not applicable.

**Informed Consent Statement:** Not applicable.

**Data Availability Statement:** Data are available on request due to privacy.

**Acknowledgments:** The authors acknowledge support from the Beijing Municipal Science and Technology Project (Grant number Z201100008020008). The authors also acknowledge support from the National Key Research and Development Project (Grant number 2021YFB3901302).

**Conflicts of Interest:** The authors declare no conflict of interest. The funders had no role in the design of the study; in the collection, analyses, or interpretation of data; in the writing of the manuscript; or in the decision to publish the results.

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
