# Peer review of "Lidar-Based 3D Obstacle Detection Using Focal Voxel R-CNN for Farmland Environment"

_agronomy, doi:10.3390/agronomy13030650_

Round 1

Reviewer 1 Report

The authors propose a modification of Voxel R-CNN called Focal Voxel R-CNN to assist the segmentation of tractors and pedestrians from point clouds and 2D images. The overall of manuscript is well-written and only minor revisions are required.

1. There're some misspellings such as at Line 124 (acquisition), Line 362, and Line 364 (=equals).

2. The authors should clarify more about the parameter inside each equation such as equation 5, what is c and each f ?

3. In table 1, what is the occlusion rate in this case and how to calculate it?

4. At Line 470, is it possible to show the accuracy of each class detection so the readers can get more in-depth information of each model?  

Reviewer 2 Report

Authors have proposed lidar based 3D object obstacle detection using focal voxel RCNN for farmland environments. The manuscript is well written and the work is interesting. I just have few comments to further improve the manuscript. 

Sectionwise outline is missing. It should added at the end of the introduction section. 

An overview diagram of the the proposed work would be meaningful and helpful for the readers of this work. 

I found only one paper 2022 is cited. I suggest the authors to cite more recent paper and some early access papers from 2023 also. The following literature is one of my suggestions "Deep Learning-Based Leaf Disease Detection in Crops Using Images for Agricultural Applications"

Hyperparameter tuning details of the deep learning model should be presented. 

Conclusion should be supported by results.